# Metabolomic and Gene Expression Studies Reveal the Diversity, Distribution and Spatial Regulation of the Specialized Metabolism of Yacón (*Smallanthus sonchifolius*, Asteraceae)

**DOI:** 10.3390/ijms21124555

**Published:** 2020-06-26

**Authors:** Guillermo F. Padilla-González, Evelyn Amrehn, Maximilian Frey, Javier Gómez-Zeledón, Alevtina Kaa, Fernando B. Da Costa, Otmar Spring

**Affiliations:** 1AsterBioChem Research Team, Laboratory of Pharmacognosy, School of Pharmaceutical Sciences of Ribeirão Preto, University of São Paulo, Av do café s/n, 14040-903 Ribeirão Preto, SP, Brazil; febcosta@fcfrp.usp.br; 2Jodrell Laboratory, Royal Botanic Gardens, Kew, Kew Green Road, London TW9 3AB, UK; 3Department of Biochemistry of Plant Secondary Metabolism, Institute of Biology, University of Hohenheim, Garbenstraße 30, 70599 Stuttgart, BW, Germany; E.Amrehn@uni-hohenheim.de (E.A.); maximilian_frey@uni-hohenheim.de (M.F.); javier.gomez@uni-hohenheim.de (J.G.-Z.); alevtinakaa@web.de (A.K.); o.spring@uni-hohenheim.de (O.S.)

**Keywords:** *Smallanthus sonchifolius*, Asteraceae, LC-MS/MS, metabolomics, gene expression, chalcone synthase, germacrene A oxidase

## Abstract

*Smallanthus sonchifolius*, also known as yacón, is an Andean crop species commercialized for its nutraceutical and medicinal properties. The tuberous roots of yacón accumulate a diverse array of probiotic and bioactive metabolites including fructooligosaccharides and caffeic acid esters. However, the metabolic diversity of yacón remains unexplored, including the site of biosynthesis and accumulation of key metabolite classes. We report herein a multidisciplinary approach involving metabolomics, gene expression and scanning electron microscopy, to provide a comprehensive analysis of the diversity, distribution and spatial regulation of the specialized metabolism in yacón. Our results demonstrate that different metabolic fingerprints and gene expression patterns characterize specific tissues, organs and cultivars of yacón. Manual inspection of mass spectrometry data and molecular networking allowed the tentative identification of 71 metabolites, including undescribed structural analogues of known bioactive compounds. Imaging by scanning electron microscopy revealed the presence of a new type of glandular trichome in yacón bracts, with a distinctive metabolite profile. Furthermore, the high concentration of sesquiterpene lactones in capitate glandular trichomes and the restricted presence of certain flavonoids and caffeic acid esters in underground organs and internal tissues suggests that these metabolites could be involved in protective and ecological functions. This study demonstrates that individual organs and tissues make specific contributions to the highly diverse and specialized metabolome of yacón, which is proving to be a reservoir of previously undescribed molecules of potential significance in human health.

## 1. Introduction

*Smallanthus sonchifolius* (Poepp. et Endl.) H. Robinson (Asteraceae), also known as yacón, is a perennial hemicryptophyte tuberous shrub native to the South American Andes, widely cultivated for its nutraceutical and medicinal properties [1,2]. The tuberous roots of yacón accumulate significant amounts of fructooligosaccharides of the inulin type, which promote health by modulation of the gut microbiome, increased mineral absorption and reduced blood glucose levels [3,4,5]. For instance, several studies have linked the consumption of yacón with improved parameters related to type II diabetes, such as α-glucosidase and α-amylase inhibitory activities [6,7], improved insulin production [8] and decreased blood glucose levels [9,10,11], among others [12,13]. The anti-inflammatory [14], antimicrobial [15,16] and antioxidant [17,18,19] activities of organic and aqueous extracts of yacón tubers and aerial parts have also been described. Such a diversity of biological effects has been attributed to the diversified specialized metabolism (SM) of yacón. Previous studies have demonstrated that the SM of species of *Smallanthus* is characterized by the presence of hundreds of compounds across several chemical classes, including terpenoids (especially diterpenes and sesquiterpene lactones—STLs), *trans*-cinnamic acid derivatives (CADs, mainly caffeic acid esters), flavonoids and lignans, among others [19,20,21,22,23,24,25,26,27,28]. In yacón, the chemical diversity of STLs, diterpenes, flavonoids, and caffeic acid esters is especially remarkable [28]. 

Due to its widespread use as a food and nutraceutical, most studies have focused on describing the chemical composition and biological properties of yacón tuberous roots. Consequently, the metabolic diversity in other plant parts, such as the flowers, stems, fibrous roots and leaves (to a lesser extent), has been poorly explored. Furthermore, previous studies suggest that different parts of yacón accumulate metabolites from different chemical classes or skeleton types, which could potentially lead to different biological effects. For example, fructooligosaccharides and caffeic acid esters are frequently reported in the tuberous roots of yacón, which is the organ attributed to the nutraceutical, anti-diabetic and antioxidant activities reported hitherto [13,23,24,29]. On the other hand, the leaves of this species accumulate significant amounts of STLs and acyclic diterpenes, while *ent*-kaurene-type diterpenes have been described in both organs: leaves and tubers. These compounds exert antimicrobial and anti-inflammatory properties [14,16,30,31]. Similar to the tubers, hypoglycemic, antioxidant and other anti-diabetes-related effects have also been described for the leaves [8,10,32]. Despite awareness of the metabolic differences, the identity and diversity of specialized metabolites across the different organs of yacón has not yet been properly characterized. Furthermore, there are no studies exploring the relationship between the expression levels of key biosynthetic genes and the accumulation patterns of the specialized metabolites in different organs of yacón. Since chemical phenotypes are the final product of genomic differences or differential gene expression, studies exploring organ-specific metabolic differences and its relationship with gene expression data can provide further insights into the distribution and spatial regulation of the SM in this medicinally important crop. 

Previous studies have described the presence of approximately 200 different metabolites in *S. sonchifolius*. However, despite the numerous reports of specialized metabolites in this species and others in the genus *Smallanthus*, most studies are based on classic phytochemical approaches. The classical process of isolation and structural elucidation is still an inherently slow and largely manual process that often leads to the identification of compounds already known to occur in a given species. Furthermore, due to their inherent limitations in terms of coverage and time, classic approaches are not suitable for large-scale studies aiming to explore the specialized metabolite diversity of an organism or tissue. Thus, global comparisons of chemical compositions present in different tissues, organs and cultivars of yacón can only be achieved through metabolomics [28,33,34,35]. Despite recent advances, the annotation, classification and identification of the plant metabolome is still a major bottleneck in all metabolomics studies. However, in recent years, a series of mass spectrometry-based computational tools such as in-silico fragmentation predictors [36,37] and molecular networking [38,39] combined with spectral library matching, substructure recognition topic modeling [40] and in silico network annotation propagation [41] have dramatically enhanced the annotation of specialized metabolites [42,43].

Following a metabolomics and gene expression approach, we recently showed the influence of environmental factors and plant developmental stage on the SM of yacón [28]. In that study, we demonstrated that the metabolic diversity of yacón leaves increases with the plant age, while environmental factors such as solar radiation and temperature induce a fast response in gene expression and an increased accumulation of flavonoids and caffeic acid esters in yacón leaves. However, the metabolic diversity in other organs remains unexplored, including the site of biosynthesis and accumulation of key metabolite classes. Therefore, by using modern analytical platforms and computational techniques, we aimed to: (1) reveal the specialized metabolite diversity and distribution in different organs, inner tissues and external appendages of two yacón cultivars and (2) correlate metabolic fingerprints with the expression patterns of key genes involved in the biosynthesis of two of the main classes of specialized metabolites in yacón: flavonoids (chalcone synthase—CHS) and STLs (germacrene A oxidase—GAO). This is especially relevant for the general knowledge of the species and for the future exploitation of the horticulturally value of yacón as a source of compounds with potential nutraceutical and medicinal applications. 

## 2. Results and Discussion

### 2.1. Metabolic Fingerprinting of Different Organs and Cultivars

Individuals from two yacón cultivars (red and white) were cultivated in field conditions from April to October 2017. Different organs, including the leaves (from different ages), stems, rhizomes and bracts, were collected from three individuals of each cultivar and extracted with 70% ethanol. The metabolic fingerprints of those samples were recorded by ultra-high-performance liquid chromatography coupled to UV detection and high-resolution tandem mass spectrometry (UHPLC-UV-HRMS/MS) to afford 2339 and 1638 mass signals in the positive and negative ionization modes, respectively. A principal component analysis (PCA) of the positive ion mode dataset grouped plant organs according to the similarities of their metabolic fingerprints. This analysis revealed a clustering tendency by plant organ in which similar organs, even from different cultivars, displayed similar metabolic fingerprints (Figure 1). Interestingly, the bract (WB), although ontogenetically a leaf, clustered clearly separately from the other organs, suggesting significant metabolic differences.

Analysis of the same dataset by orthogonal projections to latent structures discriminant analysis (OPLS-DA) (R^2^ = 0.99, Q^2^ = 0.89) and concomitant dereplication revealed a differential accumulation of metabolites by plant part (Table 1). Compounds from different chemical classes, including flavonoids, diterpenes, *trans*-cinnamic acid derivatives (CADs) and sesquiterpene lactones (STLs) constitute the main discriminant substances found in the leaves of yacón (Table 1). According to the OPLS-DA model, smaditerpenic acid F, quercetin-3-*O*-galactoside and a dicaffeoylaltraric acid isomer represent the main biomarkers of yacón leaves. Quercetin-3-*O*-galactoside was unambiguously identified by retention time (Rt) and high-resolution mass (HRMS) comparisons with a reference substance, while the identity of smaditerpenic acid F and dicaffeoylaltraric acid was tentatively suggested based on database comparisons (in the Dictionary of Natural Products and the AsterDB database) and by interpretation of MS/MS data (Appendix A). Both compounds have been previously reported in *Smallanthus sonchifolius* [24,31]. A similar approach was used for the tentative identification of raffinose, the main biomarker found in the stems of yacón, and for a trisaccharide which, along with a tricaffeoylaltraric acid isomer, correspond to the main discriminant substances found in yacón roots (Table 1). To gain a better insight into the identity of this tricaffeoylaltraric acid isomer, a sample of yacón roots was extracted with methanol 80% and submitted to classic isolation processes using Sephadex LH20 column chromatography (see Section 3.6). Comparison of the NMR spectra of the isolated metabolite agree with the previously published values for 2,3,5/2,4,5-tricaffeoylaltraric acid [24]. Interestingly, although this metabolite has been previously reported in yacón roots, our dereplication approach suggested the presence of two additional tricaffeoylaltraric acid isomers that require further studies to characterize their structure (Appendix A). The independent clustering of the bract seen in the PCA scores plot (Figure 1) can be related to the presence of hexenyl-*O*-arabinoglucoside, a compound previously reported in yacón leaves [35,44], and other structurally related molecules accumulated in high quantities in this organ (Table 1). Significant differences between the leaves and bracts in terms of trichomes, metabolite patterns and gene expression have also been observed in sunflower (*Helianthus annus* L.) [45].

To further explore the chemical diversity of yacón and its accumulation patterns in different organs, UHPLC-UV-HRMS/MS data from the positive and negative ionization modes were submitted to molecular networking and heatmaps. Visual inspection of molecular networks from both ionization modes (Appendix A) showed that compounds from the same chemical class tend to cluster together allowing the annotation of numerous metabolites with known chemical structures as well as to propose the identity of a few potentially new structural analogues (Figure 2). Spectral library annotation and manual inspection of MS/MS data allowed the annotation of 71 metabolites (Appendix A) belonging to different chemical classes: free glycosides, organic acids, CADs (mainly caffeoylquinic- and caffeoylaltraric acids and analogues), flavonoids (free, glycosylated and anthocyanins), STLs, diterpenes and hexenyl-*O*-glycoside analogues. Detailed spectral data used in the identification of metabolites is reported in Appendix A.

Analysis of the molecular network resulting from the negative ion mode dataset (Figure 2) showed that CADs clustered the higher number of nodes, suggesting a high structural diversity of this chemical class especially in the leaves and roots of yacón (Figure 2). Among the caffeoylaltraric acids and analogues, four mono-, three di- and three tricaffeoylaltraric acid isomers were detected mainly in the leaves of yacón (Appendix A). In addition to the mono-, di- and tricaffeoylaltraric acids previously reported in this species [24,35], molecular networking and manual inspection of MS/MS data allowed proposing the tentative identity of five structural analogues (compounds **15–17**, **28** and **29**, Figure 2 and Figure 3), including one coumaroyl, one feruloyl and two trihydroxycinnamoyl derivatives with different accumulation patterns. Compound **16**, accumulated only in the leaves (Figure 3), was proposed as dicaffeoylaltraric acid monoethyl ester based on the interpretation of its fragmentation pattern (Appendix A). This compound showed a parent ion at [M−H]^−^ 561.12498 *m*/*z* and a similar fragmentation to dicaffeoylaltraric acid characterized by two consecutive neutral losses of 162.031 Da consistent with two caffeoyl units. However, contrary to dicaffeoylaltraric acid, this metabolite showed a base peak at 237.06146 *m*/*z* (Appendix A, Appendix A), consistent with an ionized molecule of altraric acid monoethyl ester or its di-methoxylated isomers. However, the lack of fragment ions representing radical losses of methyl units (−15 Da), characteristic of methoxylated molecules, suggests that a single hydroxyl group of altraric acid is esterified to an ethyl unit rather than to two hydroxyl groups esterified to two methyl groups. This hypothesis is further supported by the presence of a fragment ion at 191.01921 *m*/*z*, resulting from a neutral loss of an ethanol unit. The possibility of being an artifact was excluded by a target search of this peak using methanol as extraction solvent. Compounds **15**, **17**, **28** and **29** showed similar fragmentation patterns characterized by one or two neutral losses of caffeoyl units and an intense fragment ion at 209.030 *m*/*z* representing an ionized molecule of altraric acid (Appendix A). However, compounds **28** and **29**, accumulated only in the roots (Figure 3), showed neutral losses consistent with two caffeoyl and one coumaroyl or feruloyl unit, respectively, and were therefore suggested as dicaffeoylcoumaroylaltraric acid (compound **28**, Appendix A) and dicaffeoylferuloylaltraric acid (compound **29**, Appendix A). Compounds **15** and **17** showed both a neutral loss of 178.026 Da, consistent with a trihydroxycinnamoyl unit, and were proposed as trihydroxycinnamoyl analogues of di- and tri-caffeoylaltraric acid, respectively (Appendix A). Although the approach of combining molecular networking and detailed analysis of MS/MS data allows hypothesizing about the identity of previously undescribed metabolites, the isolation and structural elucidation of these metabolites is still necessary to confirm their identity. As none of these structural analogues have been reported to date, the isolation and structural elucidation of these metabolites is currently being undertaken. 

Similar to the *trans*-cinnamic acid derivatives, glycosylated flavonoids showed a high structural diversity (Figure 2 and Appendix A). This chemical class was accumulated mainly in the leaves of yacón. However, some metabolites, such as rutin, methoxyquercetin-7-*O*-rutinoside and kaempferol-3-*O*-hexoside were accumulated also in the stems (Compounds **39**, **43** and **44**, Figure 3), while dimethoxyquercetin-7-*O*-hexoside and trimethoxygossypetin-3-*O*-acetylhexoside were detected only in the roots (compounds **51** and **52**, respectively, Figure 3). Most of the annotated flavonoid glycosides (Figure 2) represent new reports for the genus *Smallanthus* and one of them (compound **52**, Figure 3, suggested to be trimethoxygossypetin-3-*O*-acetylhexoside) might even represent a previously undescribed metabolite. This compound showed a parent ion at [M−H]− 563.14148 *m*/*z* and fragment ions at 503.12051 *m*/*z* and 359.07751 *m*/*z* (Appendix A, Appendix A), consistent with a neural loss of an acetyl unit and an acetylhexoside unit, respectively. Interestingly, the presence of an intense peak at 358.06998 *m*/*z*, representing the radical aglycone ion, suggest the glycosylation occurs at position 3 of the flavonol backbone (Appendix A) [47]. Furthermore, the three consecutive losses of 15 Da from the aglycone peak suggest the presence of three methoxy groups. However, further investigations are still necessary to unambiguously characterize the chemical structure of this metabolite. Considering that the biosynthesis of flavonoids occurs mainly in the leaves of yacón (see Section 2.2), the reason for the restricted accumulation of this metabolite in the roots requires further studies (see Section 2.3 for a plausible explanation).

Molecular networks resulting from the analysis of the positive ion mode dataset (Figure 4) allowed the annotation of several STLs and diterpenes (Appendix A) previously reported in yacón [26,31]. The annotation of hexenyl-*O*-arabinoglucoside and its structural analogues along with eight anthocyanins was also suggested by the analysis of the positive ion mode dataset. Among STLs, nine metabolites were annotated based on their clustering patterns in the molecular networks and by manual inspection of the raw spectral data (Appendix A). Most of the annotated STLs belong to the melampolide structural subtype, which represents the chemical class most commonly reported in the genus *Smallanthus* [30,48]. STLs were accumulated mainly in leaves of both yacón cultivars, although minor amounts of enhydrin, the main STL found in yacón, were also detected in the stems and bracts (Figure 3 and Figure 4). In addition to the STLs, three acyclic diterpenes, namely smaditerpenic acids C, E and F, were detected in the leaves of both yacón cultivars (Figure 3). Acyclic diterpenes constitute a common class of specialized metabolites well-described in yacón and previous studies suggest they are preferentially accumulated in the capitate glandular trichomes of yacón [31]. 

Among the glycosylated metabolites, detailed analyses of MS/MS data and spectral comparisons with literature information allowed the annotation of henexyl-*O*-araboniglucoside and two structural analogues (benzyl-*O*-araboniglucoside and phenylethyl-*O*-araboniglucoside, Appendix A), representing potentially new reports for the genus. Henexyl-3-*O*-araboniglucoside has been recently reported as a minor constituent of yacón leaves [35,44], and previous studies suggest it possess ecological implications as an anti-herbivory agent [49]. Therefore, the accumulation of henexyl-*O*-arabinoglucoside in high quantities in the bracts of yacón (Figure 3) could be related to a specialized mechanism to protect the reproductive organs from herbivory. However, additional analyses are still necessary to test this hypothesis. Among the annotated anthocyanins, two compounds, namely cyanidin-*O*-rutinoside and peonidin-*O*-rutinoside, showed a restricted accumulation in the stems and leaves of the red yacón cultivar (compounds **30** and **31**, Figure 3), suggesting they must be the underlying cause of its characteristic color. Peonidin and cyanidin glycosides constitute natural pigments responsible for the characteristic red and purple colors found in aerial and underground organs of several plant groups, including members of the Asteraceae family [50] and the tubers of some potato cultivars (*Solanum tuberosum* L., Solanaceae) and other *Solanaceous* vegetables [51,52].

In addition to the restricted accumulation of cyanidin-*O*-rutinoside and peonidin-*O*-rutinoside in the red cultivar, other metabolites also served to discriminate between both yacón cultivars. Some STLs, such as enhydrin, polymatin A and polymatin B, were accumulated in relatively higher amounts in the leaves of the white cultivar, while others, such as fluctuadin and polymatin B aldehyde, were relatively higher in the leaves of the red cultivar (Figure 3). While no qualitative variability was found in STLs contents between different cultivars, our results suggest semi-quantitative variations, which are in line with previous reports [26,35]. The same tendency was observed in the accumulation of smaditerpenic acids. While smaditerpenic acid C was accumulated mainly in the leaves of the red cultivar, its two isomers, smaditerpenic acids E and F, were mainly accumulated in the white cultivar (Figure 3). To further explore metabolic differences between both yacón cultivars, an additional analysis by OPLS-DA (R^2^ = 1.00, Q^2^ = 0.99, Appendix A) was performed. This analysis revealed important differences in the metabolic composition of the leaves of both cultivars, allowing the identification of the main discriminant metabolites (Table 2). Among them, 5-*O*-(*E*)-caffeoylquinic acid, smaditerpenic acid F and a dicaffeoylaltraric acid isomer represent the main biomarkers of the white cultivar, while smaditerpenic acid C, methoxygossypetin-3-*O*-hexoside, and cyanidin-*O*-rutinoside are preferentially accumulated in the leaves of the red cultivar (Table 2).

### 2.2. Gene Expression Patterns in Different Organs and Cultivars

The expression patterns of key genes involved in the biosynthesis of flavonoids (chalcone synthase—CHS) and STLs (germacrene A oxidase—GAO) were studied in the leaves (from different ages), stems, roots and bracts of two yacón cultivars. Gene expression analysis of CHS (Figure 5) showed similar trends to the accumulation patterns of flavonoids in different organs (Figure 3). CHS was found to be mainly expressed in adult leaves, while no significant expression was observed in the roots and bracts of both yacón cultivars (Figure 5). A differential CHS expression was found in the stems of each cultivar with a high expression in the red and no expression in the white cultivar (Figure 5). Analysis of the accumulation patterns of flavonoids in different organs of yacón (Figure 3) revealed a similar trend to the CHS expression with a higher accumulation in the leaves of both cultivars. However, the stems of the red cultivar did not show a greater accumulation of flavonoids when compared to the white cultivar (Figure 3). Therefore, the higher CHS expression seen in the stems and leaves of the red cultivar might be related to the higher accumulation of anthocyanins, especially cyanidin-*O*-rutinoside and peonidin-*O*-rutinoside. The biosynthesis of anthocyanins follows the same initial steps of all flavonoids via CHS, but in addition to such enzyme, anthocyanidin synthase (ANS) occurs at later steps catalyzing the biosynthesis of anthocyanidins which are subsequently glycosylated to form anthocyanins [52,53]. Therefore, the higher accumulation of these metabolites in the red cultivar is likely a consequence of a higher CHS and ANS expression in the stems and leaves of this cultivar.

Interestingly, the fact that two glycosylated flavonoids, namely dimethoxyquercetin-7-*O*-hexoside and trimethoxygossypetin-3-*O*-acetylhexoside (compounds **51** and **52**, Figure 3), were accumulated in the roots of both cultivars while there was no CHS expression in that organ (Figure 5) suggest that downstream biosynthetic enzymes in the flavonoid pathway could be restricted to yacón roots or that specialized transportation mechanisms could be acting. Previous studies have shown that the site of biosynthesis and accumulation of specialized metabolites can differ according to the ecological roles they play in nature [54,55,56]. For example, in *Nicotiana* L. (Solanaceae) species, nicotine is biosynthesized in root cells and subsequently translocated to the leaves via xylem, where it is accumulated in the vacuoles to provide protection against insect herbivores [54]. According to Gutierrez et al. (2017), “many studies have identified specific transporters involved in the translocation of flavonoids from one organ to another”. For instance, a long-distance transport of specialized metabolites has been demonstrated in *Arabidopsis thaliana* (L.) Heynh. (Brassicaceae) [57] and the possibility of long-distance flavonoid translocation in other species has also been suggested, based on the presence of specific transporters on vascular bundles [58]. Among the specific mechanisms involved in the transport of flavonoids, the presence of ATP-binding cassette (ABC)- and multidrug and toxic compound extrusion (MATE)-type transporter proteins, as well as glutathione S-transferases, are predominantly cited [55,56,59]. Additionally, it has been suggested that depending on the substituent group of the flavonoid moiety, a complex vesicle trafficking system including the Golgi apparatus can also be involved [59]. 

Contrary to CHS, the expression of GAO in different organs of yacón (Figure 5) did not follow the accumulation patterns of STLs seen in the heatmap (Figure 3). GAO was found to be highly expressed in young leaves (RYL, WYL), but no significant expression was found in adult leaves (RL, WL), stems and roots of both cultivars (Figure 5). Although adult leaves did not show a significant expression of GAO, this organ accumulated the highest quantity of STLs (Figure 3), which agrees with recent literature reports comparing the STLs profiles of yacón leaves over time [28]. According to Göpfert et al. (2005), the biosynthesis of STLs occurs in specific organs and time points inside the stalk cells of capitate glandular trichomes (CGT) present in leaf primordia and young leaves at active secretory stages. In old leaves, the secretory activity is already concluded and STLs are stored in the epicuticular globe of CGT [60]. This developmental regulation of STL biosynthesis would explain the contrasting trend observed between the expression level of GAO and the accumulation patterns of STLs [28]. The relatively low GAO expression seen in yacón bracts (Figure 5) is supported by the low accumulation of STLs found in this organ (Figure 3). Interestingly, most of the STLs found in yacón leaves and bracts have a melampolide backbone hydroxylated in positions C8 and C14 (Figure 5A). Thus, with the sequence information of other sesquiterpene-related genes, such as HaG8H, HaES and HaC14H (CYP71AX36) [61] and the definition of the secretory stage of the glandular trichomes in yacón, the elucidation of new metabolic pathways of STL will be possible in the future for this species. The inherent substrate plasticity of GAO demonstrated for a wide range of sesquiterpene backbones suggests that this enzyme is responsible for the remarkable diversification of STLs, especially in members of the Asteraceae family [62].

### 2.3. Accumulation Patterns of Specialized Metabolites in Specific Organs and Tissues

To further study the accumulation patterns of the main classes derived from the SM in yacón (CADs, flavonoids and STLs), we investigated characteristic metabolites from these main classes in external appendages of aerial parts, as well as in inner tissues and underground organs. Initially, we used scanning electron microscopy to elucidate the main types of trichomes present in the leaves and bracts of yacón, while UHPLC-UV-HRMS/MS was used to characterize the chemical composition of manually collected trichomes. Microscopic analysis of yacón leaves showed the presence of three main types of trichomes: one biseriate capitate glandular type (CGT) and two uniseriate non-glandular linear trichomes (LT and sLT, Figure 6) of different size and shape, which are in accordance with previous reports [63,64,65]. The second type of linear trichomes found in this study (sLT, Figure 6) is consistent with the flexuous type previously described by Mercado et al. (2006). Although a previous paper described an additional type of trichomes in the ligules of the ray flowers of yacón [65], our results suggest that such trichomes, described as “two-celled uniseriate non-glandular trichomes with bulbous base and rounded apex” are comparable to the immature stage of the “LT” trichomes previously described. Linear trichomes of different developmental stages are usually found growing nearby in the flowers of this species. Interestingly, analysis of yacón bracts showed a previously undescribed type of trichome, hereafter referred to as multiseriate capitate glandular trichomes (MCGT, Figure 6), structurally similar to the biseriate CGT type found on the leaves but composed of a multiseriate stem and a multicellular head from which the cuticular globe is formed (Figure 6). This type of trichome, considerably larger than the CGT type (Figure 6), showed a highly dense accumulation on the bracts of yacón, while they were absent on the leaves or stems of the same species. The scarce flowering of this species in some localities and the fact that most studies have focused on the leaves to describe the trichomes’ morphology of yacón could be the main reason the MCGT type has not been reported previously in this widely used medicinal species. 

UHPLC-UV-HRMS/MS analyses of manually collected CGT from yacón leaves showed that this type of trichomes accumulates high quantities of STLs and acyclic diterpenes (Figure 7), which is in accordance with previous reports [26,31,66,67]. Previous studies have reported that CGT constitute the main biosynthetic and accumulation site of STLs in several species of Asteraceae, such as *Helianthus annuus* [45,60], *Tanacetum parthenium* L. [68], *Aldama discolor* (Baker) E.E.Schill. & Panero [69] and *Tithonia diversifolia* (Hemsl.) A.Gray [70]. In yacón, CGT are well known to accumulate STLs and acyclic diterpenes where more than 20 different metabolites from both chemical classes have been identified [26,31]. Based on detailed analyses of fragmentation patterns and HRMS comparisons with published spectra, we found six major STLs and two acyclic diterpenes, among which enhydrin and smaditerpenic acid F constitute the main metabolites (Figure 7). Analysis of the MCGT showed that this unusual type of trichome also accumulates STLs and acyclic diterpenes (Figure 7) but at a different relative proportion. Contrary to the observed trend in the CGT, we found that the accumulation of smaditerpenic acid F in MCGT was considerably higher compared to the accumulation of enhydrin and other STLs in this type of trichome (Figure 7). Lastly, analysis of LT demonstrated the presence of several unidentified and structurally related molecules requiring further investigation. 

Analysis of the distribution patterns of characteristic metabolites of the flavonoid and CADs pathways in inner tissues and underground organs of yacón (Figure 8) showed that while most flavonoids are accumulated in the leaves (Figure 3), significant amounts of a select few (i.e., rutin and quercetin-3-*O*-galactoside, Figure 8) were also detected in the stem epidermis, where they could be involved in protection mechanisms against UV-B radiation. Several studies have reported that high levels of flavonoids are accumulated in the upper epidermal layer of plant cells in response to UV-B radiation [71,72]. Thus, the fact that those flavonoids were not detected in the inner layers of the stem nor in the fibrous or tuberous roots is consistent with this hypothesis. On the other hand, the restricted presence of dimethoxyquercetin-7-*O*-hexoside and trimethoxygossypetin-3-*O*-acetylhexoside in the roots of yacón and their absence in the leaves (Figure 3) and internal parts of the stems and tubers (Figure 8) generates more questions on their possible ecological roles. Previous studies have suggested that the accumulation of flavonoids in underground organs serves as root–rhizosphere signaling molecules involved in the establishment of mycorrhizal associations, as chemoattractors of rhizobia towards the root, as inhibitors of root pathogens or as mediators of allelopathic interactions, among others [73]. The accumulation of caffeic acid esters in high quantities in the root’s and tuber’s epidermis of yacón agrees with previous studies reporting the high antioxidant potential of crude extracts from those organs [24,74]. However, the current study suggests that yacón fibrous roots accumulate relatively higher amounts of caffeic acid esters compared to the tuberous roots (Figure 8).

Regarding the accumulation of STLs, although they are the major specialized metabolites of CGT, the presence of trace quantities in the inner tissues of the stem and in the roots of yacón remains elusive. Previous studies on sunflower hypocotyls demonstrate that minor amounts of specific STLs (from a different subclass to the STLs found on the leaves) are accumulated in inner tissues where they play important physiological roles [75,76]. For example, STLs are thought to be involved in plant phototropism, as some of them inhibit auxins (plant-growth hormones) and are accumulated in a light-dependent manner in the early stages of plant development (e.g., sunflower hypocotyls) causing the bending of the plant in the direction of light [75,76]. According to Yokotani-Tomita et al. (1997, 1999), 8-epixanthatin is an auxin-inhibiting STL accumulated in higher concentrations (up to three times) in unilateral illuminated sunflower hypocotyls compared to the shaded side. The phototropic curvature in sunflower hypocotyls may therefore be caused by a lateral gradient of the STL 8-epixanthatin. On the other hand, the presence of minor amounts of STLs in sunflower root exudates has been linked with the ecological roles they play as stimulatory substances of plant growth [77] or in the established of symbiotic associations with arbuscular mycorrhiza fungi (reviewed in [78]). Arbuscular mycorrhizal associations have also been described for yacón and its wild relative *S. macroscyphus* [79], but it is currently unknown whether SM play a role in these symbiotic associations. Previous reports have demonstrated that sunflower root exudates containing STLs can induce the growth of different plant species, including obligate plant parasites [77], while structural analogues of STLs (strigolactones) can promote the hyphal branching of arbuscular mycorrhizal fungi at very low concentrations [80,81].

## 3. Materials and Methods 

### 3.1. Plant Material

Individuals of *Smallanthus sonchifolius* from two different cultivars were grown on a trial field of the University of Hohenheim (Stuttgart, Germany) for six months (April to October 2017) until they reached the adult stage. Plant collections were performed from three individuals of each cultivar in the same date and time of the day. Each cultivar came from a different clone held at the Botanic Garden of the University of Hohenheim, propagated by rhizome cuttings in the form of germinated rhizomes. Cultivars were differentiated by the color of their tubers’ peels and stems: named *S. sonchifolius* cv red and *S. sonchifolius* cv. white, in accordance with the literature [1,74]. After cleaning with distilled water, different organs, including the leaves, aerial stems, fibrous roots and bracts, were collected and stored at −70 °C until further use. Adult leaves of three different ages were also collected: the first fully expanded leaf (young leaves, L1), a leaf in the middle of the stem (adult leaves, L2) and the last non-senescent leaf (old leaves, L3). Samples of the stems were collected from the middle part of the stem. Bracts were collected only from the “white” cultivar, as this was the only one that produced flowers. 

For further analysis of the distribution patterns of metabolites in external appendages and internal tissues, the epidermis of the tuberous roots and stems of the “white” yacón cultivar were separated from the inner tissues (vascular bundles) with a scalpel and tweezers under anatomical lens and analyzed by UHPLC-UV-HRMS/MS. 

The chemical composition of the different types of trichomes present in yacón leaves and bracts was also investigated. Trichomes were manually collected (approximately 100) using the technique of glandular trichome microsampling [82,83] in 2 mL Eppendorf tubes filled with 500 µL of HPLC-grade acetonitrile and analyzed by UHPLC-UV-HRMS/MS. 

Three individuals from each cultivar were collected and treated as biological replicates. For metabolomic analyses, biological replicates were pooled according to their tissue, cultivar and plant organ before the extraction of metabolites. For gene expression studies, biological replicates were extracted independently.

### 3.2. Microscopic Analyses

Scanning electron microscopy was used to study the morphology of different types of trichomes present in yacón leaves and bracts. For this, a DSM 940 scanning microscope (Zeiss, Jena, Germany) was used following the method previously described in [84,85,86]. Briefly, freshly harvested leaves and bracts were fixed with FAA fixative (formaldehyde: glacial acetic acid: 70% alcohol, 5:5:90, by vol) for 24 h. After cell fixation, samples were dehydrated at room temperature (RT) in 70% acetone for 30 min, 100% acetone for 30 min and 100% acetone for 60 min, followed by a critical point drying (CPD) with a CPD 030 (Leica EM, Wetzlar, Germany). The dried samples were mounted on aluminum stubs with a double-sided carbon tape and coated with gold–palladium (80:20) using a sputter coater (SCD 040, Oerlikon Balzers, Pfäffikon, Switzerland). The investigation of the samples was carried out on a DSM 940 scanning electron microscope at an accelerating voltage of 5 kV. The micrographs were digitized with Orion, version 6.38 (Orion Microscopy, Charleroi, Belgium) and Adobe Photoshop CS6 (Adobe Systems Inc., San Jose, CA, USA) was used to adjust brightness and contrast, and for one picture the feature photo merge. All solvents used for the microscopic analyses were purchased from Carl Roth GmbH (Karlsruhe, Germany).

### 3.3. Metabolite Extraction and UHPLC-UV-HRMS/MS Analysis

Fifty milligrams (fresh weight) of each plant organ and tissue were collected in 2 mL reaction tubes and a 2.8 mm stainless steel bead was added to each of them. After 2 min in liquid nitrogen, samples were ground on a MM400 mixer mill (Retsch GmbH, Haan, Germany) for 30 s at a frequency of 30 Hz. Homogenized samples were extracted with 70% aqueous ethanol (1 mL) by vortexing for a few seconds followed by ultrasonication at room temperature for 10 min at 40 kHz. After extraction, samples were centrifuged for 5 min at 13,000 rpm and the supernatant was filtered through a 0.2 µm PTFE filter. The extraction process used in this study was based on the protocol for extraction and analysis of plant tissues for metabolomics studies [87].

Metabolic fingerprinting by UHPLC-UV-HRMS/MS was performed on an Agilent UHPLC system (Santa Clara, CA, USA) coupled to an 80 Hz photodiode array detector (PDA) and a *Q Exactive Plus* (Thermo Scientific, Waltham, MA, USA) high-resolution tandem mass spectrometer. Five microliters of each plant extract were separated on an Acquity CSH C18 column (1.7 μm, 150 mm × 2.1 mm, Waters, Milford, MA, USA) using (A) water (0.2% formic acid) and (B) acetonitrile (0.2% formic acid) as mobile phase at a flow rate of 400 μL/min. Chromatographic separation was performed under the following gradient program: 0–15 min, 3–20% B; 15–40 min, 20–95% B; 40–43 min, 95–3% B; 43–45 min, 3% B. UV detection was performed between 190 and 400 nm.

The column effluent was ionized by electrospray using a capillary temperature of 360 °C, a heater temperature of 380 °C and spray voltages of +4.2 kV and −3.5 kV for the positive and negative ionization modes, respectively. Total Ion Current chromatograms were obtained over the range of 140–1200 *m*/*z* using the *Fullscan* (resolution of 70,000 FWHM) and *data-dependent MS^2^* (dd-MS^2^, resolution of 17,500 FWHM) methods and the following parameters: automatic gain control (AGC) target, 1.0 · 10^6^ (*Fullscan*) and 5.0 · 10^4^ (dd-MS^2^); maximum injection time, 500 ms (*Fullscan*) and 64 ms (dd-MS^2^); sheath gas flow rate, 60; auxiliary gas flow rate, 20; topN for dd-MS^2^, 5 and an isolation window of 1.5 *m*/*z*. Nitrogen was used as the drying, nebulizer and fragmentation gas.

### 3.4. Data Preprocessing and Multivariate Analyses

Raw chromatographic data obtained for the positive ionization mode was uploaded to the software MZMine 2.28 (Okinawa, Japan and Espoo, Finland) [88] where several preprocessing steps were performed, including peak detection, peak filtering, chromatogram construction, chromatogram deconvolution, isotopic peak grouping, chromatogram alignment, gap filling, duplicate peaks filter, and fragment and adduct search. To separate compound signals from instrumental or chemical noise, we employed the exact mass algorithm considering a value of 1.0 · 10^5^ as noise threshold. A peak resolution of 70,000 FWHM was considered along with the Lorentzian extended peak model function to remove shoulder peaks resulting from the Fourier transformation. Peaks were detected using the chromatogram builder function with the following parameters: 0.2 min as minimum time span, 5.0 · 10^5^ as minimum height and a *m*/*z* tolerance of 0.002. Chromatogram deconvolution was performed using the XCMS wavelets algorithm, 0.25 to 5.0 wavelets scales and a S/N threshold of 10. Isotopes were removed using the isotopic peaks grouper function considering a retention time tolerance of 0.7 min and a *m*/*z* tolerance of 0.002. Ransac alignment was employed as a peak alignment algorithm using a threshold value of 10 and 0 interactions and to allow the software to calculate the adequate number of interactions. Peaks not detected after alignment due to low intensity or poor quality were detected using the peak finder function as a gap filling algorithm, considering an intensity tolerance of 30%, a mass tolerance of 0.002 *m*/*z* and a retention time tolerance of 0.3 min. Duplicate peaks were removed using the duplicate peaks filter, a mass tolerance of 0.002 *m*/*z* and a retention time tolerance of 0.3 min. Fragment peaks and common adducts observed in each ionization mode were identified using the same *m*/*z* and retention time tolerance previously mentioned. 

The data matrix of the positive ionization mode obtained after MZmine preprocessing was edited as an Excel spreadsheet (Microsoft Windows, Redmond, WA, USA), where peaks detected in the blank sample (extraction solvent) and in the last minutes of the chromatographic run were excluded before the statistical analyses. Multivariate analyses were performed in the software R 3.0.3 (R Foundation for Statistical Computing, Vienna, Austria) and SIMCA 13.0.3.0 (Umetrics, Umeå, Sweden), considering Pareto as scaling method. Initially, PCA was performed as an exploratory method to determine the clustering tendency of the samples based on the similarities of their metabolic fingerprints. To identify discriminant metabolites, OPLS-DA was performed considering PCA groups as Y variable. Lastly, to have an overview of the relative accumulation of the identified metabolites (see below) in the studied samples, we performed an analysis by heatmaps in the R package *gplots.* LC-MS/MS data supporting the metabolomic experiments are publicly available as a MassIVE dataset (https://gnps.ucsd.edu) under the following accession code: MSV000085101.

### 3.5. Dereplication of Plant Extracts

Molecular networking based on MS/MS spectral similarity was employed as dereplication strategy of plant extracts. This method allows organizing “large datasets of tandem mass spectra based on the similarity between fragmentation patterns of different, but related, precursor ions” [38]. As a key advantage, molecular networking makes it possible to perform metabolite annotation of unknown mass signals by spectral matching with reference substances available in the GNPS database [39]. This analysis was performed following the online workflow described at the GNPS website (https://gnps.ucsd.edu/ProteoSAFe/static/gnps-splash.jsp). Chromatographic data in raw format from the positive and negative ionization modes were transformed to .mzXML format using the MSConvert package from the software ProteoWizard 3.0.9798 (Proteowizard Software Foundation, Palo Alto, CA, USA). Consensus MS/MS spectra were created via MS-Cluster by grouping repeatedly acquired spectra from the same molecules to achieve a higher signal-to-noise ratio, under the following parameters: parent and fragment ion mass tolerance of 0.02 Da and a minimum of two MS/MS spectra to be considered for molecular networking. Consensus spectra from different molecules (represented as nodes) were connected by edges when they shared a minimum of 4 common fragment ions and a cosine score above 0.65. Edges between two nodes were kept in the network only if both nodes were within the top 10 most similar nodes to each other. To perform spectral library annotation, we considered a mass tolerance of 0.02 Da and a minimum of four common fragment ions between our experimental data and the spectral library of reference substances. Lastly, the software Cytoscape 3.6.1 (Seattle, WA, USA) [89] was used to edit and display the final structure of the network.

To confirm and expand the spectral library annotation made by molecular networking, accurate mass values, MS/MS fragmentation patterns, UV spectra and retention time values of the detected metabolites were manually inspected and compared with literature data and authentic standards analyzed under identical experimental conditions. The following reference substances were used: quinic acid, 5-*O*-(*E*)-caffeoylquinic acid, 3,4-di-*O*-(*E*)-caffeoylquinic acid, 3,5-di-*O*-(*E*)-caffeoylquinic acid, 4,5-di-*O*-(*E*)-caffeoylquinic acid, rutin, quercetin-3-*O*-galactoside, enhydrin, uvedalin and longipilin acetate. Comparisons of accurate mass values with data from the literature were performed relative to the monoisotopic masses (<5 ppm accuracy) of the secondary metabolites reported in the Dictionary of Natural Products (DNP, http://dnp.chemnetbase.com), in SciFinder Scholar (https://scifinder.cas.org) and in our *in*-house Asteraceae Database (AsterDB, www.asterbiochem.org/asterdb), the last of which contains more than 2500 metabolites reported in the family Asteraceae, including several members of the genus *Smallanthus*. To have an overview of the confidence level achieved in the identification of metabolites, we adopted the four levels of accuracy reported in the Metabolomics Standard Initiative [46]. Experimental data, including retention times, accurate mass values, fragment ions and the accuracy level achieved in the identification of metabolites are reported in Appendix A.

### 3.6. Isolation of Tricaffeoylaltraric Acid

Considering that chromatographic analyses indicated the presence of three putative tricaffeoylaltraric acid isomers with different accumulation patterns in different organs of yacón, a sample of yacón roots was submitted to classic isolation processes aiming to structurally characterize one of those isomers and to validate our dereplication approach. Briefly, approximately 30 g of yacón roots (dry weight) were ground and extracted initially with dichloromethane and subsequently with methanol 80% in two consecutive steps with each solvent for 24 h. The methanol extract was dried under reduced pressure and 4 g of it was submitted to Sephadex LH-20 column chromatography (30 cm × 2.5 cm i.d.) employing mixtures of water–ethanol (100:0 to 0:100) to produce 12 fractions (Fr1-Fr12). UHPLC-UV-MS/MS monitoring of the obtained fractions showed that fraction 12 (4 mg) was constituted by pure tricaffeoylaltraric acid. The structure of the isolated metabolite was identified by uni- and bidimensional NMR experiments (DMSO-d6, Bruker ARX 400, Billerica, MA, USA) and by high resolution MS and MS/MS.

### 3.7. RNA Extraction and RT-qPCR

Total RNA from fresh adult leaves, stems, roots and bracts was extracted from plant tissues (20 mg) collected in 2 mL reaction tubes. Considering that previous studies have shown a higher expression of key genes involved in the SM in young leaves and leaf primordia of closely related species [45,85,90], we included young developing leaves (3 days old) in the gene expression studies. Metabolomic comparisons between young developing leaves and adult leaves is published elsewhere (Padilla-Gonzalez et al., 2019). After tissue collection, a 2.8 mm stainless steel bead was added to each reaction tube, and tubes were immediately placed in liquid nitrogen. Samples were subsequently ground on a MM400 mixer mill (Retsch GmbH) for 30 s at a frequency of 30 Hz. RNA was extracted using the EURx GeneMATRIX Universal RNA Purification kit (Roboklon, Berlin, Germany), following the manufacturer’s instructions with the following modification: forty milligrams of insoluble PVP was added after the RL buffer to absorb polyphenols. RNA quality and quantity were verified in a BioPhotometer (Eppendorf, Hamburg, Germany). After RNA isolation, samples were confirmed to be free from gDNA by a PCR analysis performed in a peqSTAR Thermocycler (Peqlab, Erlangen, Germany), using RNase/DNase-free water, RedTaq MasterMix (Genaxxon Bioscience, Ulm, Germany) and actin primers (Table 3). The following program was used in the PCR amplification: 4 min of initial denaturation followed by 38 cycles of 15 s denaturation (96 °C), 15 s annealing (54 °C), 1 min · kb-1 elongation (72 °C) and final elongation for 4 min (72 °C). In case of positive results for residual gDNA, it was eliminated with the Perfecta Dnase I (Rnase-free) kit (Quanta Biosciences, Beverly, MA, USA) before cDNA synthesis. The RNA concentration of all samples was then adjusted to 50 ng/mL and cDNA was synthesized using the RevertAid cDNA Synthesis Kit (Thermo Scientific), using the primer VNdT18-Oligonucleotide at a final concentration of 5 µM. 

Real time qPCR analyses were performed in a CFX96 Touch™ qPCR System (BioRad, Hercules, CA, USA) using SYBR Green I for visualization (SensiFast^TM^ SYBR No-ROX Kit, Bioline, London, UK) and the software Bio-Rad CFX Maestro 1.0 (Bio-Rad). The expression patterns of key genes involved in the biosynthesis of flavonoids (chalcone synthase, CHS) and STLs (germacrene A oxidase, GAO) in yacón [28] were investigated in different plant organs (Table 3). Three housekeeper genes were selected in accordance with literature reports: actin (ACT), elongation factor (EF) and glycerin aldehyde phosphate dehydrogenase (GAPDH) [28,90]. However, based on the stability values shown by the three housekeeper genes in different organs of yacón (calculated according to [91]), only the actin and elongation factor primers (ACT = 0.005 and EF = 0.007) were used to normalize the CHS and GAO expression in the qPCR analyses. HPLC-purified qPCR primers were used in a final concentration of 0.25 μM, under the following program: 2 min at 95 °C for initial denaturation, followed by 45 cycles of 15 s at 95 °C and 15 s at 60 °C. A melting curve analysis following each RT-qPCR was performed to assess product specificity (Appendix A), and the efficiency of all primers was calculated using four serial dilutions of 1:5 (Appendix A). The nucleotide sequence, amplicon length, annealing temperature and efficiency of the investigated primers are reported in Table 3. Gene expression patterns were analyzed in three biological replicates, each with three technical replicates.

## 4. Conclusions

By following a multidisciplinary approach, this study provides a comprehensive analysis of the spatial regulation, diversity and distribution of specialized metabolites in yacón. Specifically, our results suggest that different metabolic fingerprints characterize specific tissues, organs and cultivars of yacón, while our molecular networking-based approach allowed the annotation of 71 metabolites from different chemical classes, including six potentially new compounds: five caffeoylaltraric acid analogues and one flavonoid. Furthermore, the expression patterns of chalcone synthase (CHS) and germacrene A oxidase (GAO) showed significant differences according to the plant organ and cultivar. Although both genes are mainly expressed in the leaves, our results suggest that gene expression depends on the developmental stage of this organ. CHS was mainly expressed in adult leaves, while GAO showed a higher expression in immature leaves. The fact that high quantities of the flavonoids dimethoxyquercetin-7-*O*-hexoside and trimethoxygossypetin-3-*O*-acetylhexoside were found in yacón fibrous roots, while adult leaves constitute the main biosynthetic site for this chemical class, implies that specialized mechanisms could be acting to transport certain flavonoids from their place of biosynthesis (leaves) to the roots or the restricted presence of downstream biosynthetic enzymes in yacón roots. Lastly, scanning electron microscopy and chromatographic analyses were used to describe the chemical composition of different types of trichomes, inner tissues of aerial stems and underground organs and to discuss their possible physiological and ecological roles. Therefore, despite its well-known nutritional and medicinal properties, the specialized metabolite diversity of yacón is still proving to be a reservoir of previously undescribed compounds of potential significance in human health and can be used in future studies as a model to further understand ecological roles of specialized metabolites.

## Figures and Tables

**Figure 1 ijms-21-04555-f001:**
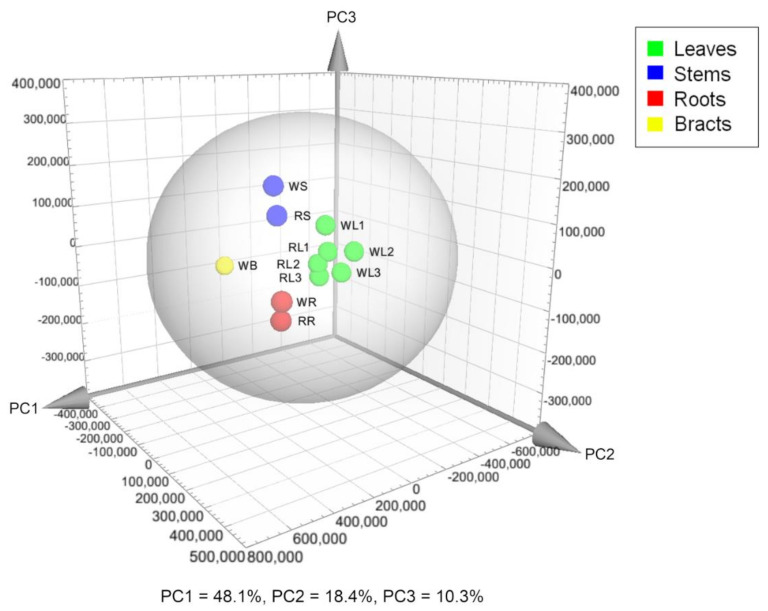
Principal component analysis scores plot based on metabolic fingerprinting by UHPLC-UV-HRMS in positive ion mode of different organs of yacón. Plant organ abbreviated with different letters (L: leaves; S: stems; R: roots; B: bracts) and yacón cultivar abbreviated with the letters W: “white” and R: “red”. Leaves from three different ages were considered: the first fully expanded leaf (young leaves, L1), a leaf in the middle of the stem (adult leaves, L2) and the last non-senescent leaf (old leaves, L3). Each sphere represents mean values from three biological replicates pooled according to their cultivar and plant organ before the extraction of metabolites.

**Figure 2 ijms-21-04555-f002:**
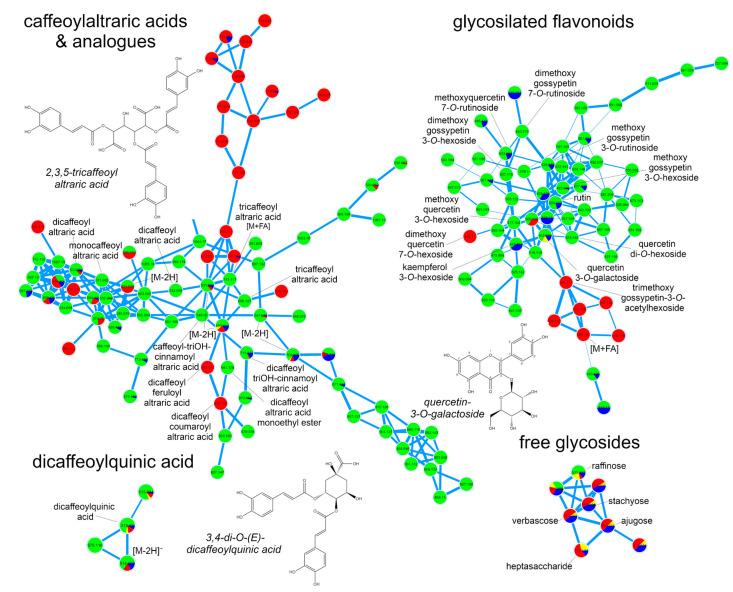
Molecular networking of yacón extracts based on metabolic fingerprinting by UHPLC-UV-HRMS/MS in negative ion mode. Node colors represent different organs (green: leaves; red: roots; blue: stems and yellow: bracts).

**Figure 3 ijms-21-04555-f003:**
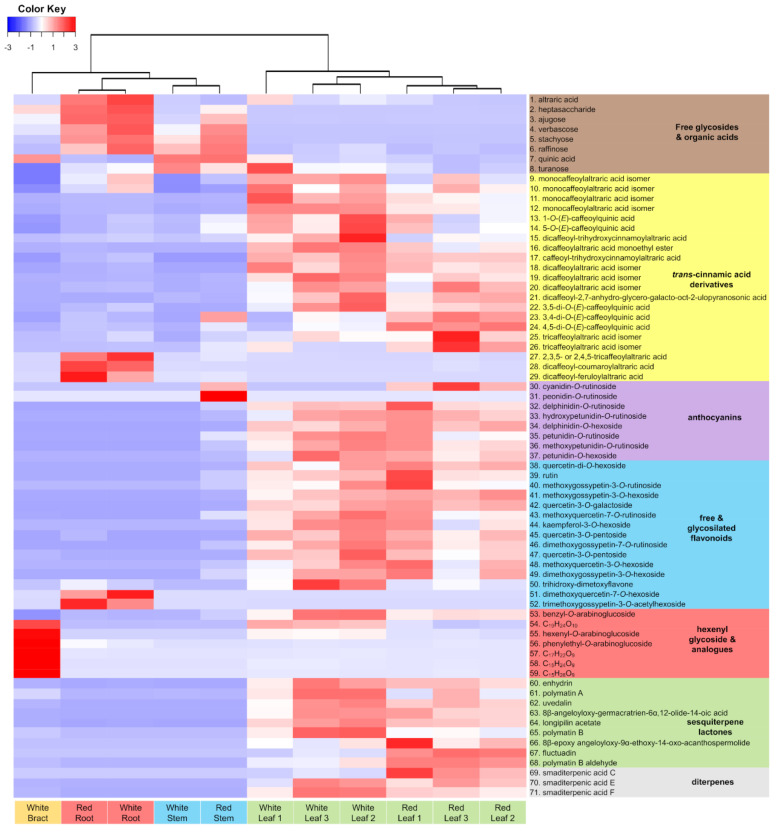
Heatmap showing the differential accumulation of 71 metabolites in extracts from different organs of yacón analyzed by UHPLC-UV-HRMS. Compounds were identified by Rt, HRMS and MS/MS comparisons with reference substances and by interpretation of MS/MS data and database information available in the Dictionary of Natural Products, AsterDB and SciFinder Scholar. Complete mass spectrometry information and confidence level achieved in the identification of metabolites is available in Appendix A.

**Figure 4 ijms-21-04555-f004:**
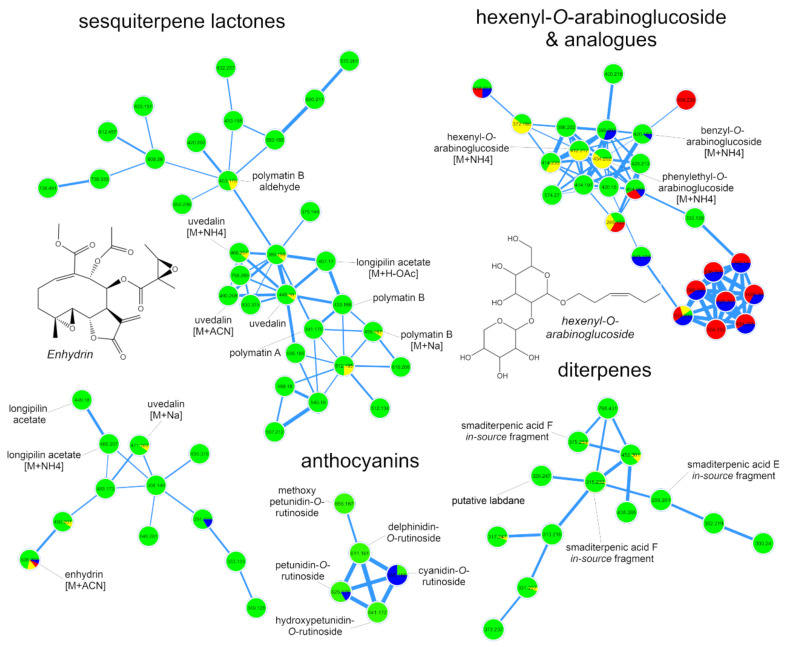
Molecular networking of yacón extracts based on metabolic fingerprinting by UHPLC-UV-HRMS/MS in positive ion mode. Node colors represent different organs (green: leaves; red: roots; blue: stems and yellow: bracts).

**Figure 5 ijms-21-04555-f005:**
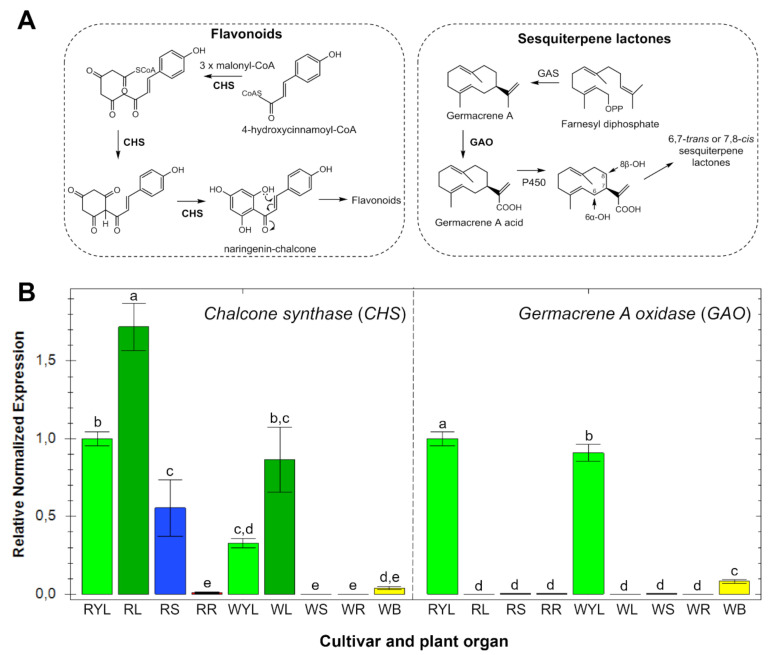
(**A**) biosynthetic pathways of flavonoids and STLs in Asteraceae and (**B**) gene expression patterns of chalcone synthase (CHS) and germacrene A oxidase (GAO) in different organs of yacón. Bars colored according to the plant organ (light green: young leaves (YL); dark green: old leaves (L); blue: stems (S); red: roots (R) and yellow: bracts (B)). White and red yacón cultivars distinguished by the letters W and R, respectively. Same letters into the bar graphs do not differ statistically. Confidence intervals represent deviation in the expression values from three biological replicates.

**Figure 6 ijms-21-04555-f006:**
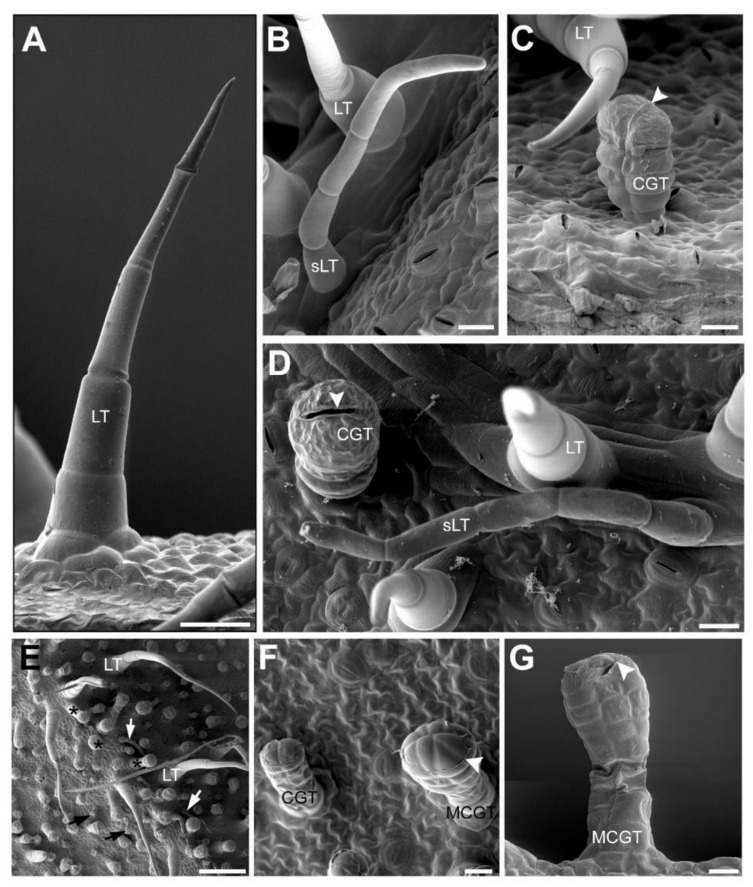
Scanning electron micrographs showing the morphology of different types of trichomes of *Smallanthus sonchifolius*: (**A**–**D**) trichome types of leaves; (**E**–**G**) trichome types of bracts; (**A**) non-glandular uniseriate linear trichome (LT) with cells which were thinning from the base to the tip, scale bare: 50 µm; (**B**) small uniseriate trichome (sLT) with cells of similar shape, scale bare: 20 µm; (**C**) capitate biseriate glandular trichome (CGT) with a cuticle globe showing a predetermined breaking point (white arrowhead) in closed state, scale bare: 20 µm; (**D**) orientation of the three types of trichomes on the leaf. CGT and LT had a vertically orientation and sLT were often orientated horizontally to the leaf surface. The white arrowhead marks the predetermined breaking point at the beginning of rupture, scale bare: 20 µm. This picture is a photo merge of two micrographs; (**E**) overview of trichome types of bracts with some LT, CGT (black arrows), sLT (white arrows) and several multiseriate capitate glandular trichomes (black asterisks) which were exclusively found on bracts, scale bare: 200 µm; (**F**) top view of a biseriate (CGT) and a multiseriate capitate glandular trichome (MCGT) showing the different number of tip cells, white arrow head marked the destroyed cuticle globe which was formed by the multicellular head; scale bare: 20 µm; (**G**) profile of a MCGT, white arrow head—destroyed cuticle globe, scale bare: 20 µm.

**Figure 7 ijms-21-04555-f007:**
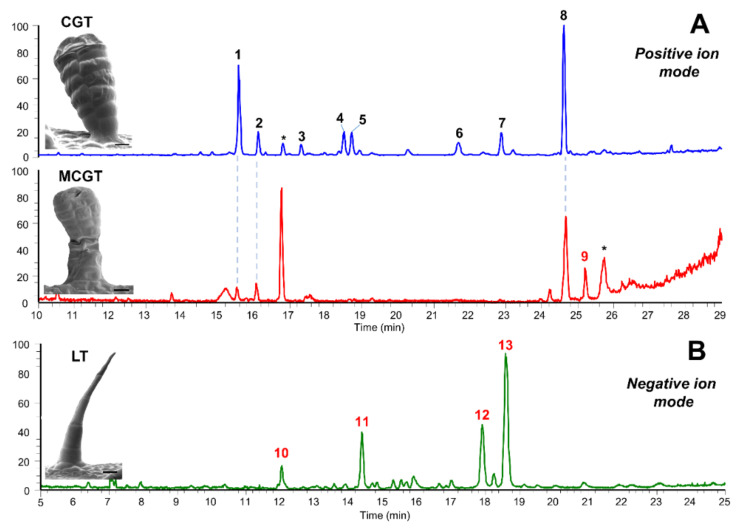
Scanning electron micrographs and total ion current chromatograms of manually collected trichomes. (**A**) Biseriate capitate glandular trichomes (CGT, scale bare: 10 µm) collected from yacón leaves and multiseriate capitate glandular trichomes (MCGT, scale bare: 20 µm) from yacón bracts analyzed in the positive ion mode. (**B**) Non-glandular uniseriate linear trichomes (LT, scale bare: 50 µm) analyzed in the negative ion mode. Peak numbers represent the metabolites identities as follows: 1. enhydrin, 2. polymatin A, 3. uvedalin, 4. (1Z,4E)-8ß-angeloyloxy-germacra-1(10),4,11(13)-trien-6a,12-olide-14-oic acid, 5. longipilin acetate, 6. polymatin B, 7. smaditerpenic acid E, 8. smaditerpenic acid F, 9. C_25_H_34_O_4_, 10. C_23_H_30_O_7_, 11. C_23_H_28_O_6_, 12. C_25_H_30_O_7_, 13. C_25_H_30_O_7_. * solvent-derived peaks.

**Figure 8 ijms-21-04555-f008:**
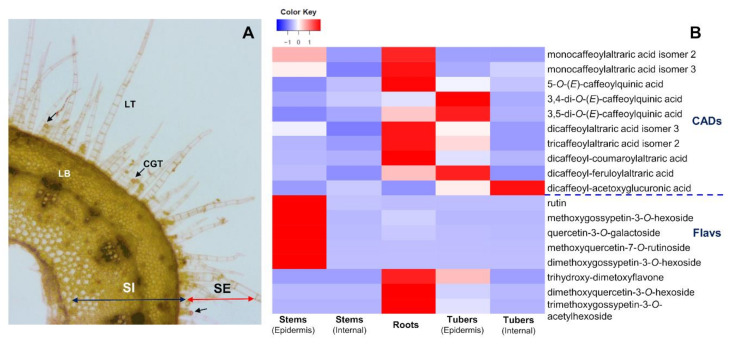
Distribution patterns of flavonoids and CADs in the fibrous roots, epidermis and inner tissues of the stems and tubers of yacón. (**A**) Light micrograph of a transversal cut of yacón stems showing linear trichomes (LT), capitate glandular trichomes (CGT, black arrows) and vascular tissues (LB). (**B**) Heatmap showing the accumulation patterns of specific metabolites in each tissue.

**Table 1 ijms-21-04555-t001:** Discriminant metabolites of different organs of yacón identified by the OPLS-DA model. Metabolites sorted by their variable importance in the projection (VIP) value (higher to lower).

Rt	*m*/*z*	[*pseudo*-Molecular Ion] → MS^2^ Ions	Discriminant Substance (Comments)	Confidence *
**Leaves**				
24.55	315.232	[M+H−H_2_O]^+^ 375.253→330.894, 315.233bp, 269.223	smaditerpenic acid F (*in-source* fragment)	2
8.37	303.050	[M+H]^+^ 465.103→303.049bp, 285.038, 257.045, 229.050	quercetin-3-*O*-galactoside (*in-source* fragment)	1
8.26	163.039	[M+H]^+^ 535.108→163.039bp	dicaffeoylaltraric acid (*in-source* fragment)	3
8.43	333.060	[M+H]^+^ 495.113→333.060bp 318.037	methoxygossypetin-3-*O*-hexoside (*in-source* fragment)	2
8.43	495.113	[M+H]^+^ 495.113→333.060bp 318.037	methoxygossypetin-3-*O*-hexoside	2
8.37	465.103	[M+H]^+^ 465.103→303.049bp, 285.038, 257.045, 229.050	quercetin-3-*O*-galactoside	1
15.40	482.202	[M+H]^+^ 465.175→405.154, 349.128, 289.107bp, 229.086	enhydrin (NH_4_^+^ adduct)	1
**Stems**				
1.26	381.079	[M+Na]^+^ 527.258→365.106bp, 347.095, 203.053, 185.042	raffinose (*in-source* fragment)	3
1.26	527.158	[M+Na]^+^ 527.258→365.106bp, 347.095, 203.053, 185.042	raffinose (Na^+^ adduct)	3
1.26	543.132	[M+Na]^+^ 527.258→365.106bp, 347.095, 203.053, 185.042	raffinose (K^+^ adduct)	3
**Roots**				
1.31	504.192	[M+NH_4_]^+^ 504.1192→163.059, 145.049bp, 127.039, 97.029, 85.029	trisaccharide (NH_4_^+^ adduct)	3
16.05	441.048	[M+H]^+^ 441.048→361.092bp, 346.068, 329.066, 301.071, 167.034	C_21_H_12_O_11_	4
11.28	163.039	[M+H]^+^ 697.139→163.039bp	2,3,5/2,4,5-tricaffeoylaltraric acid (*in-source* fragment)	1
**Bracts**				
8.55	412.217	[M+ NH_4_]^+^ 412.217→145.050, 127.039, 115.039, 97.029, 91.058bp, 85.029, 73.029	hexenyl-*O*-arabinoglucoside (NH_4_^+^ adduct)	2
10.86	366.175	[M+ NH_4_]^+^ 366.175→145.049, 127.039, 105.019, 97.029, 85.029bp	C_15_H_24_O_9_ (NH_4_^+^ adduct)	4
12.57	430.171	[M+ NH_4_]^+^ 430.170→145.049, 127.039, 109.029, 105.019, 97.029, 85.029bp	C_19_H_24_O_10_ (NH_4_^+^ adduct)	4
10.64	388.158	[M+ NH_4_]^+^ 388.158→145.049, 127.039, 105.019, 97.029, 85.029bp	C_17_H_22_O_9_ (NH_4_^+^ adduct)	4
12.12	368.191	[M+ NH_4_]^+^ 368.191→145.049, 127.039, 105.019, 97.029, 85.029bp	C_15_H_26_O_9_ (NH_4_^+^ adduct)	4

* According to the metabolomics standards initiative [46]. Level 1: identified by Rt, HRMS and MS/MS comparisons with a reference substance; Level 2: identified by database comparisons of HRMS data and by interpretation of UV spectra and fragmentation patterns; Level 3: chemical class suggested by HRMS comparisons with online databases, UV data and chemotaxonomic information; Level 4: mass features with unknown identity, or for which the obtained data were not conclusive. bp: base peak.

**Table 2 ijms-21-04555-t002:** Discriminant metabolites of different cultivars of yacón sorted by their variable importance in the projection (VIP) value (higher to lower).

Rt	*m*/*z*	[*pseudo*-Molecular Ion] → MS2 Ions	Discriminant Substance (Comments)	Confidence *
**White cultivar**				
5.96	163.039	[M+H]^+^ 355.102→163.039bp	5-*O*-(*E*)-caffeoylquinic acid (*in-source* fragment)	1
1.22	286.092	[M+H]^+^ 286.092→124.039bp, 85.029	C_12_H_15_O_7_N	4
8.26	163.039	[M+H]^+^ 535.108→163.039bp	dicaffeoylaltraric acid isomer (*in-source* fragment)	3
24.61	315.233	[M+H−H_2_O]^+^ 375.253→330.894, 315.233bp, 269.223	smaditerpenic acid F (*in-source* fragment)	2
11.12	430.171	[M+ NH_4_]^+^ 430.171→145.049, 127.039bp, 105.019, 97.029, 85.029	C_19_H_24_O_10_ (NH_4_^+^ adduct)	4
5.96	377.084	[M+H]^+^ 355.102→163.039bp	5-*O*-(*E*)-caffeoylquinic acid (Na^+^ adduct)	1
5.96	355.102	[M+H]^+^ 355.102→163.039bp	5-*O*-(*E*)-caffeoylquinic acid	1
**Red cultivar**				
19.17	315.232	[M+H−H_2_O]^+^ 333.242→315.232bp, 297.22098, 269.22659	smaditerpenic acid C (*in-source* fragment)	2
8.43	495.113	[M+H]^+^ 495.113→333.060bp 318.037	methoxygossypetin-3-*O*-hexoside	2
4.29	449.108	[M]+ 595.166→449.108, 287.055bp	cyanidin-*O*-rutinoside (*in-source* fragment)	2
10.98	478.265	[M+H]^+^ 461.238→443.228, 417.208, 213.185, 127.039bp	C_22_H_36_O_10_ (NH_4_^+^ adduct)	4
19.17	373.235	[M+H−H_2_O]^+^ 333.242→315.232bp, 297.22098, 269.22659	smaditerpenic acid C (Na^+^ adduct)	2
8.43	333.060	[M+H]^+^ 495.113→333.060bp 318.037	methoxygossypetin-3-*O*-hexoside (*in-source* fragment)	2

* According to the metabolomics standards initiative [46]. Level 1: identified by Rt, HRMS and MS/MS comparisons with a reference substance; Level 2: identified by database comparisons of HRMS data and by interpretation of UV spectra and fragmentation patterns; Level 3: chemical class suggested by HRMS comparisons with online databases, UV data and chemotaxonomic information; Level 4: mass features with unknown identity, or for which the obtained data were not conclusive. bp: base peak.

**Table 3 ijms-21-04555-t003:** Primer combinations used in PCR and qPCR. Germacrene A oxidase (Ss_qGAO), Chalcone synthase (Ss_qCHS), Actin (Ha_qACT), Elongation Factor (Ha_EF), and glycerin aldehyde phosphate dehydrogenase (Ha_qGAPDH). Housekeeper genes originally designed for *Helianthus annus* (abbreviated as Ha).

Code	Primer (5′ - 3′)	Size (bp)	T_a_ (°C)	Efficiency	R^2^
Ss_qGAO	F: CGAAAACGGCAACACCACCATT	162	60	92.3	0.998
	R: GCTCGCACCATTGGGAAGTTTC				
Ss_qCHS	F: GCCGACTACCAGCTCACCAAACTC	193	60	87.2	0.999
	R: CCTCATTAGGGCCACGGAACG				
Ha_qACT	F: GCCGTGCTTTCTCTTTATGCCAGCGACC	137	60	95.0	0.998
	R: AGCGAGATCAAGACGAAG				
Ha_qEF	F: ACCAAATCAATGAGCCCAAGAGACCCA	131	60	99.6	0.987
	R: TACCGGGCTTGATCACACCAG				
Ha_qGAPDH	F: GCAAGGACTGGAGAGGTGGAAGAG	140	60	92.6	0.998
	R: ATCAACGGTAGGGACACGGAATG				

T_a_: annealing temperature.

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
