# Peer review of "Metabolomic and Gene Expression Studies Reveal the Diversity, Distribution and Spatial Regulation of the Specialized Metabolism of Yacón (Smallanthus sonchifolius, Asteraceae)"

_ijms, 2020, doi:10.3390/ijms21124555_

Round 1

Reviewer 1 Report

Dear Editor/Author

Most studies about yacón (Smallanthus sonchifolius) concluded that due to functional properties, yacon roots may be used as a dietary supplement to prevent and treat diseases. Therefore, the subject of the work seems to be inspiring not only because of the research aspects of the work, but also because of its practical application. All experiences have been carefully planned by authors and developed in detail. Various analytical techniques were used to verify the assumptions. I wanted to find a weak job point but it's really good.

I recommended this paper to publication in International Journal of Molecular Sciences as “accept in present form”.

Author Response

We would like to thank this reviewer for the time taken to assess our manuscript and for the encouraging words

Reviewer 2 Report

Comments to the Author

This manuscript describes a very interesting approach to metabolomics and gene expression studies reveal the diversity, distribution and spatial regulation of the secondary metabolites in yacón. The idea involves a study of metabolic diversity using metabolomics and gene expression in yacón plant parts, such as the flowers, stems, roots and leaves. I found the concept interesting, the work clearly described, and the conclusions unambiguous. The main issue then is the novelty of the work.  I have little hesitation recommending acceptance, because the fundamental idea behind this work had already been published by same authors.  The prior description of the basic idea makes the present manuscript seem like a follow-up with being extending previous work, at its core, simply an extension of previous work using same methods of analysis. Therefore, I am able to recommend this work for the publication in this journal after modification of introduction, so that introduction covers more information about authors’ previous published work.

Author Response

We would like to thank the reviewer for time taken to assess our manuscript and for the valuable suggestion. We completely agree that information about our previous publication should be mentioned in more detail in the introduction and we thank for pointing it out. As suggested, we have included the following paragraph in lines 89-95: "Following a metabolomics and gene expression approach, we recently showed the influence of environmental factors and plant developmental stage on the SM of yacón [28]. In that study, we demonstrated that the metabolic diversity of yacón leaves increases with the plant age, while environmental factors such as solar radiation and temperature induce a fast response in gene expression and an increased accumulation of flavonoids and caffeic acid esters in yacón leaves. However, the metabolic diversity in other organs remains unexplored, including the site of biosynthesis and accumulation of key metabolite classes."